# Adaptive Textual Label Noise Learning based on Pre-trained Models

**Shaohuan Cheng,  Wenyu Chen,  Mingsheng Fu,  Xuanting Xie,  Hong Qu**[*]
School of Computer Science and Engineering,
University of Electronic Science and Technology of China
shaohuancheng@std.uestc.edu.cn, cwy@uestc.edu.cn, fms@uestc.edu.cn
x624361380@outlook.com, hongqu@uestc.edu.cn

## Abstract

The label noise in real-world scenarios is unpredictable and can even be a mixture of different types of noise. To meet this challenge, we develop an adaptive textual label noise learning framework based on pre-trained models, which consists of an adaptive warm-up stage followed by a hybrid training stage. Specifically, an early stopping method, relying solely on the training set, is designed to dynamically terminate the warm-up process based on the model's fit level to different noise scenarios. The hybrid training stage incorporates several generalization strategies to gradually correct mislabeled instances, thereby making better use of noisy data. Experiments on multiple datasets demonstrate that our approach performs on-par with or even better than the state-of-the-art methods in various noise scenarios, including scenarios with the mixture of multiple types of noise.

## 1 Introduction

In recent years, deep neural networks (DNNs) have been successfully applied in many fields (Pouyanfar et al., 2018; Alinejad et al., 2021; Liu et al., 2023) and the performance largely depends on well-labeled data. However, accessing large-scale datasets with expert annotation in the real world is difficult due to the significant time and labor costs involved. Instead, the noisy data obtained directly from the real world is often utilized in practical scenarios, even though it inevitably contains some incorrect labels. Thus, research on learning with noisy labels has gained attention in various fields such as natural language processing (NLP) (Jindal et al., 2019; Jin et al., 2021; Wu et al., 2022).

There are two main types of label noise in NLP: class-conditional noise (CCN) and instance-dependent noise (IDN). CCN assumes that label noise is dependent on the true class, which can simulate the confusion between similar classes like

"Game" and "Entertainment". On the other hand, IDN assumes that label noise is dependent on the instance, which simulates the confusion caused by the characteristics of the instance. For example, a piece of news containing the phrase "played on a pitch as slow as a bank queue" may be misclassified as Economic news instead of Sports news due to the specific wording. However, most studies focus on a particular type of noise. For example, Jindal et al. (2019) introduces a non-linear processing layer to learn the noise transition matrix of CCN. Qiao et al. (2022) designs the class-regularization loss according to the characteristic of IDN. However, there is a premise for applying these methods, which is that the noise is known and of a single type.

In the real-world, the noise scenarios are more complex and involve a mixture of multiple noises arising from various factors, such as data ambiguity, collection errors, or annotator inexperience. Methods that specifically target one type of noise are less effective when dealing with other types of noise. This limitation hinders their applicability in real scenarios where the noise is unknown and variable.

To address the challenges posed by real noise scenarios, we develop an adaptive textual label noise learning framework based on pre-trained models. This framework can handle various noise scenarios well, including different types of noise and mixed noise types. Specifically, our approach begins with an adaptive warm-up stage, then divides the data into clean and noisy sets by the correctness statistic of samples, and utilizes different generalization strategies on them. In particular, there are three key designs in our approach. **First**, the warm-up stage is designed to automatically stop early based on the model's fit level to the noise scenario, which effectively prevents the model overfitting erroneous labels especially under IDN scenarios or with a high ratio of noise. No-

---

[*]Corresponding author

tably, the adaptive warm-up method relies solely on the raw training set, rather than a clean validation set, making it more suitable for practical scenarios. **Second**, unlike previous works (Li et al., 2020; Qiao et al., 2022) that fit GMM (Permuter et al., 2006) on the training losses to separate data, the data is separated according to the correctness statistic of each sample. The correctness statistic is accumulated by assessing the consistency between the model's predictions and the given labels during the warm-up stage. This prolonged observation provides more accurate grounds for data partitioning. **Third**, a linear decay fusion strategy is designed to gradually correct the potential wrong labels to generate more accurate pseudo-labels by adjusting the fusion weights of the original labels and the model outputs.

We conduct extensive experiments on four classification datasets, considering different types of noise: class-conditional noise, instance-dependent noise, and a mixture of multiple noises. To the best of our knowledge, previous methods have not explored such mixed noise. The experimental results demonstrate that our method surpasses existing general methods and approaches the performance of methods specifically designed for particular noise types in different noise settings. Our contributions can be concluded as follows:

- We design an early stopping method for fine-tuning the pre-trained models, which adapts to various noise scenarios and datasets and achieves near the best test accuracy by relying solely on training set.

- We develop an adaptive noise learning framework based on pre-trained models, which can make good use of different types of noisy data while effectively preventing the model from overfitting erroneous labels.

- Experimental results of various noise settings show that our approach performs comparably or even surpasses the state-of-the-art methods in various noise scenarios, which proves the superiority of our proposed method in practical scenarios.

## 2 Related work

**Universal Label Noise Learning.** Label noise learning methods can be divided into two groups: loss correction and sample selection methods

(Liang et al., 2022). Loss correction tries to reduce the effect of noisy labels during training by adding regularization item in loss, designing robust network structure for noisy label and so on. For example, Wang et al. (2019) adds a reverse cross-entropy term to the traditional loss to reduce the disturbance brought by noise. ELR (Liu et al., 2020) adds a regularization term prevent the model from memorizing the noisy labels because it would not be fitted in the early training stage. Sample selection divides the data into clean and noisy subsets, and uses different methods for different subsets. For example, Co-Teaching (Han et al., 2018) maintains two networks. During training, the two networks respectively pick out some small-loss samples as clean data for each other to learn. DivideMix (Li et al., 2020) uses Gaussian mixture model to separate clean and noisy samples. The noisy samples are treated as unlabeled data, whose pseudo-labels are generated by the model. Finally, Mixmatch (Berthelot et al., 2019) method is adopted for mixed training on clean set and noisy set.

**Labels Noise Learning in NLP.** The above works mainly focus on vision tasks, the researches on textual scenarios are relatively fewer. Jindal et al. (2019) and Garg et al. (2021) add additional noise modules based on lightweight models such as CNN and LSTM to learn the probability of noise transfer. (Liu et al., 2022; Tänzer et al., 2021; Zhu et al., 2022; Qiao et al., 2022) conduct research on pre-trained models and find that pre-trained models demonstrate superior performance in noise learning compared to trained-from-scratch models. However, most works focus on one certain type of noise such as CCN. Qiao et al. (2022) studies both CCN and IDN, but still conducts experiments in settings where the type of noise is known and designs specific regularization loss for IDN.

Few works have focused on general methods of label noise in NLP. Zhou and Chen (2021) develops a general denoising framework for information retrieval tasks, which reduces the effect of noise by adding a regularization loss to samples whose model predictions are inconsistent with the given label. Jin et al. (2021) proposes an instance-adaptive training framework to address the problem of dataset-specific parameters and validates its versatility across multiple tasks. However, these methods rely on additional components, such as supplementary structures or auxiliary dataset, which limits their practicality. In contrast, our method relies

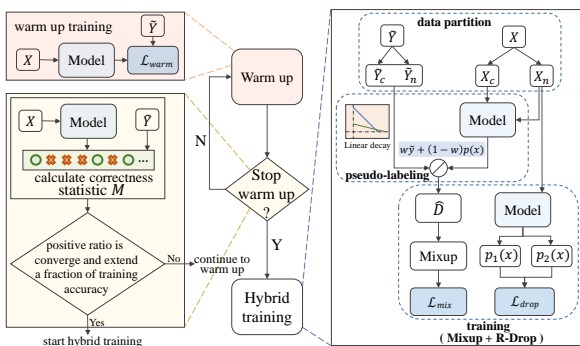

Figure 1: The overall diagram of the proposed method

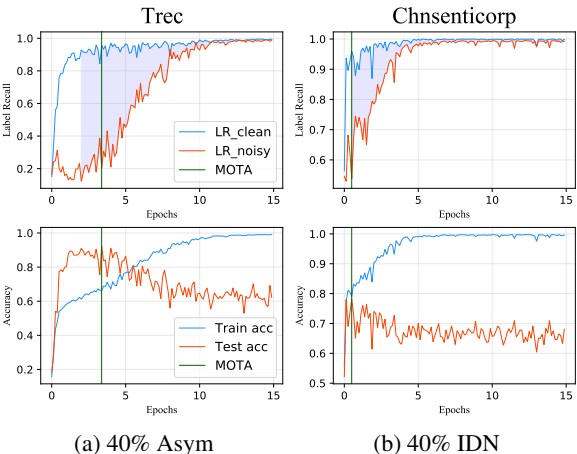

(a) 40% Asym       (b) 40% IDN

Figure 2: The memorization observation of different noise cases. Label recall on top figures, train and test accuracy on bottom figures. MOTA is short for the maximum obtainable test accuracy . More cases can be found in Appendix A.

solely on a single network and the noisy training data, making it superior in terms of practicality.

## 3  Methodology

**Problem Definition.** Without loss of generalization, we take text classification as an example. Given a noisy training dataset $\tilde{D} = (X, \tilde{Y}) = \{(x_i, \tilde{y}_i)\}_{i=1}^N$, the one-hot label $\tilde{y}_i$ associated with sample $x_i$ is probably wrong. Our goal is to learn a classification model $p(y|x; \theta, \zeta)$ from the noisy dataset $\tilde{D}$, which generalizes well on clean test data. Specifically, the classification model $p(y|x; \theta, \zeta)$ consists of a pre-trained encoder and a classifier with parameters $\theta$ and $\zeta$, respectively.

**Overview of the Proposed Method.** To address various noise types, we propose a general learning method consisting of the warm-up training stage and the hybrid training stage, and the overall diagram is shown in Figure 1. The classification model $p(y|x; \theta, \zeta)$ is first trained by the raw data in the warm-up stage to form the initial classification ability. During the warm-up stage, we also maintain a correctness statistic to justify whether the model begins to overfit the noisy data. Once there is a sign of overfitting to noisy data, we will stop the warm-up, and move on to the subsequent hybrid training stage. During the hybrid training stage, the raw data is divided into the clean set and noisy set according to the correctness statistic, and then further train model $p(y|x; \theta, \zeta)$ by applying different training strategies to the clean set and the noisy set respectively.

### 3.1  Adaptive warm-up

The goal of this warm-up stage is to obtain a classification model that fits clean data well but not noisy data. As shown in Figure 2, however, the optimal warm-up time may vary significantly for different noise scenarios. To meet this challenge,

an adaptive early stopping condition is involved to terminate the warm-up training before overfitting noisy data. The details of our warm-up stage are given as follows.

**Training Data.** We directly use the raw noisy dataset $\tilde{D}$ to warm up the classification model and determine when to stop early.

**Learning objective.** The standard cross-entropy loss is used to warm up the model:

$$\mathcal{L}_{warm} = -\sum_{i=1}^N \tilde{y}_i^T \log\big(p(\tilde{y}_i|x_i; \theta, \zeta)\big). \quad (1)$$

**Overfitting regarding noisy data.** Our stopping condition is based on the following Assumption 1 and Assumption 2.

*Assumption 1: As learning progresses, the clean data is fitted faster than the noisy data.* We empirically demonstrate that the clean samples can be recalled earlier than the noisy samples during learning (see Figure 2), regardless of the noise types. This is mainly caused by the memorization effect (Arpit et al., 2017) which means DNNs tend to learn simple patterns before fitting noise. As a result, the whole learning process can be roughly divided into Clean Rising (CR) phase (where most clean samples are quickly learned) and Noisy Rising (NR) phase (where the noisy samples are fitted slowly), which are differentiated by backgrounds in Figure 2.

*Assumption 2: The prediction regarding noisy data tends to swing between the true label and the*

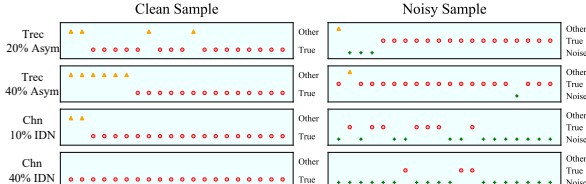

Figure 3: Examples of the model output during training under different noise settings. The given label of clean samples is true. More examples can be found in Appendix A.

*noisy label.* It originates from another memorization phenomenon regarding noisy samples(Chen et al., 2021). The results in Figure 3 also demonstrate that the prediction for the noisy samples exhibits a higher level of inconsistency with the given label due to the activation of the true label.

**Correctness statistic.** According to these two assumptions, we update the correctness statistic of training set at intervals to judge whether the model begins to overfit the noisy data. Specifically, for a given sample $x_i$ with label $\tilde{y}_i$, its correctness coefficient $m_i^t$ is:

$$m_i^t = \sum_t r_i^t, \tag{2}$$

obtained by,

$$r_i^t = \begin{cases} 1, & \text{if } \arg\max_{k \in K} p^k(x_i; \theta, \zeta) = \arg\max_{k \in K} \tilde{y}_i^k \\ -1, & \text{else.} \end{cases} \tag{3}$$

where the correctness $r_i^t$ indicates whether the prediction of sample $x_i$ is consistent with the given label at moment $t$, and $m_i^t$ is the statistic of correctness in a given time range.

The higher the number of correct predictions compared to incorrect predictions for a sample, the higher the degree to which the model fits that sample. Intuitively, positive $m_i^t$ indicates $x_i$ has been fitted by the model at moment $t$.

Furthermore, we have a proportion $ratio^t$,

$$ratio^t = \frac{1}{N} \sum_i \mathbb{I}(m_i^t > 0), \tag{4}$$

which indicates the sample fitting level for the whole dataset with $N$ samples.

**Early stopping condition.** Through observing the change of $ratio^t$, we can determine whether the learning process has entered the NR phase. According to Assumption 1, in the CR phase, $ratio^t$

should increase fast since the model fits clean samples quickly. In the NR phase, $ratio^t$ should stay within a certain range for an extended period because the noisy samples are difficult to fit (Assumption 2).

As a consequence, if $ratio^t$ stays within a range $\varepsilon$ for $\eta$ times, we can assume that the learning process has entered the NR phase. To approach the optimal stopping point, we continue to warm up the model until a certain improvement in the training accuracy. The magnitude of improvement is set to be a fraction $\rho_1$ of the current remaining accuracy. The pseudo-code for adaptive warm-up process is shown in Appendix B. Note that this adaptive warm-up process is suitable for different noise scenarios due to Assumption 1 and Assumption 2 can be widely satisfied by different noise types.

### 3.2 Hybrid training

To further leverage the underlying values of noisy samples, we propose a hybrid training method applying different training strategies to clean samples and noisy samples respectively.

**Data.** Based on the correctness statistic $M = \{m_i^{t'}\}_{i=1}^N$ (assuming $t'$ is the stopping time of the warm-up stage), the whole training set $\tilde{D} = \{(x_i, \tilde{y}_i)\}_{i=1}^N$ can be divided into the "clean" set $\tilde{D}_c$ and "noisy" set $\tilde{D}_n$ as follows.

$$\begin{aligned} \tilde{D}_c = (X_c, \tilde{Y}_c) = \{(x_i, \tilde{y}_i) | \text{if } m_i^{t'} \geq l\}, \\ \tilde{D}_n = (X_n, \tilde{Y}_n) = \{(x_i, \tilde{y}_i) | \text{if } m_i^{t'} < l\}, \end{aligned} \tag{5}$$

where $l$ is the $N\rho_2$-th largest correctness statistic value of $M$ because a larger number indicates that the sample is fitted earlier and has a higher probability of being a clean sample. $\rho_2$ is a given percentage, which is set to 20%.

**Pseudo-labeling.** To minimize the side-effects of the noisy labels, we regenerate labels for both the clean set $\tilde{D}_c$ and noisy set $\tilde{D}_n$ through the pseudo-labeling method combining the original labels and the prediction of the classification model. Note that there may inevitably be some noisy samples in set $\tilde{D}_c$, albeit fewer than in $\tilde{D}_n$.

For each sample $(x_i, \tilde{y}_i)$ in clean set $\tilde{D}_c$, the corresponding pseudo-labels $\hat{y}_i$ is obtained by,

$$\hat{y}_i = w_c^t \cdot \tilde{y}_i + (1 - w_c^t) \cdot p(x_i; \theta, \zeta), \tag{6}$$

where the weight $w_c^t$ decays linearly with the training step $t$, which is calculated by,

$$w_c^t = 1 - (1 - \delta_1) \cdot \frac{t}{T}, \tag{7}$$

where $T$ indicates the all training steps of this stage, $w_c^t$ decays from 1 to $\delta_1$ due to relatively reliable labels in the clean set $\tilde{D}_c$.

Likewise, for each sample in noisy set $\tilde{D}_n$, its pseudo-labels $\hat{y}_i$ are obtained by,

$$
\begin{aligned}
w_n^t &= \delta_2 \cdot (1 - \frac{t}{T}), \\
\hat{y}_i &= w_n^t \cdot \tilde{y}_i + (1 - w_n^t) \cdot p(x_i; \theta, \zeta),
\end{aligned}
\tag{8}
$$

where $w_n^t$ decays from $\delta_2$ to 0 due to most labels in $\tilde{D}_n$ are incorrect.

By linearly decaying the weight of original labels, a good balance is achieved between leveraging the untapped label information and discarding noise. As the accuracy of the classification model's predictions continues to improve, the generated pseudo-labels will be closer to the true labels.

Furthermore, for each pseudo-label $\hat{y}_i$ , we adopt the sharpen function to encourage the model to generate low entropy predictions, i.e., $\hat{y}_i = \hat{y}_i^{1/\tau} / \left\| \hat{y}_i^{1/\tau} \right\|_1$, where $\tau$ is the temperature parameter and $\|\cdot\|_1$ is $l_1$-norm. Finally, the new whole set $\hat{D} = \{(x_i, \hat{y}_i)\}_{i=1}^N$ is reformed by clean and noisy sets.

**Mixup.** To enhance the generalization of the model, mixup technique (Zhang et al., 2017; Berthelot et al., 2019) is adopted to introduce diverse and novel examples during training. Different from mixing pictures directly in vision tasks, the text input cannot be directly mixed due to the discreteness of words. Like previous work (Berthelot et al., 2019; Qiao et al., 2022) in NLP, the sentence presentations encoded by pre-trained encoder are used to perform mixup:

$$
\begin{aligned}
\lambda &= \text{Beta}(\alpha, \alpha) && (9) \\
\lambda' &= \max(\lambda, 1 - \lambda) && (10) \\
h' &= \lambda' p(x_i; \theta) + (1 - \lambda') p(x_j; \theta) && (11) \\
y' &= \lambda' \hat{y}_i + (1 - \lambda') \hat{y}_j && (12)
\end{aligned}
$$

where $\alpha$ is the parameter of Beta distribution, $p(x; \theta)$ is the sentence embedding which corresponds to "[CLS]" token. $(x_i, \hat{y}_i)$ and $(x_j, \hat{y}_j)$ are randomly sampled in the corrected set $\hat{D}$.

The mixed hidden states $\{h'_i\}_{i=1}^N$ and targets $\{y'_i\}_{i=1}^N$ are used to train the classifier by applying entropy loss:

$$
\mathcal{L}_{mix} = -\sum_{i=1}^N y'^T_i \log\big(p(h'; \zeta)\big)
\tag{13}
$$

| Datasets | Classes | Traning | Testing | Type |
|---|---|---|---|---|
| Trec | 6 | 5,452 | 500 | Question |
| Agnews | 4 | 120,000 | 7,600 | News Topic |
| IMDB | 2 | 25,000 | 25,000 | Movie Review |
| Chnsenticorp | 2 | 10,430 | 1,200 | Hotel Review |

Table 1: The statistics of datasets

**R-Drop.** Since the prediction of the model is fused in pseudo-labeling, it needs to be as accurate as possible. Therefore, the R-Drop strategy (liang et al., 2021) is applied on noisy samples $X_n$ to promote the model to have a consistent output distribution for the same input. R-Drop minimizes the Kullback-Leibler divergence between two distributions predicted by the model with dropout mechanism for the same sample :

$$
\begin{aligned}
\mathcal{L}_{drop} = \sum_{i=1}^M \frac{1}{2} \Big( &\mathcal{D}_{KL}\big(p_1(x_i; \theta, \zeta) || p_2(x_i; \theta, \zeta)\big) \\
+ &\mathcal{D}_{KL}\big(p_2(x_i; \theta, \zeta) || p_1(x_i; \theta, \zeta)\big) \Big),
\end{aligned}
\tag{14}
$$

where $M$ is the number of noisy samples $X_n$. $p_1(x_i; \theta)$ and $p_2(x_i; \theta)$ are two predictions of $x_i$.

**Learning objective.** The total loss is:

$$
\mathcal{L} = \mathcal{L}_{mix} + \beta \mathcal{L}_{drop},
\tag{15}
$$

where $\beta$ is the hyper-parameter of KL loss, which is set to 0.3 in experiments.

## 4 Experiments

### 4.1 Experimental settings

**Datasets** Experiments are conducted on four text classification datasets, including Trec (Voorhees et al., 1999), Agnews (Zhang et al., 2015), IMDB (Maas et al., 2011) and Chnsenticorp (Tan and Zhang, 2008), where Chnsenticorp is Chinese dataset and the rests are English datasets. The statistics of datasets are presented in Table 1, where the training set of Chnsenticorp is a merger of the original training and the validation set.

**Noise types** The following types of noise are injected to standard datasets:

**Class-conditional noise:** We choose typical symmetric (Sym) and asymmetric (Asym) noises in various class-conditional noise to conduct experiments. Symmetric noise flips a certain percentage of labels in a given category into other categories uniformly. Asymmetric noise flips labels between given similar class pairs. IMDB and Chnsenticorp

| Noise Type | | Sym | | Asym | | | | IDN | | | even mixture (Sym & Asym) | | even mixture (Asym & IDN) | | | | uneven mixture (all three) | |
|---|---|---|---|---|---|---|---|---|---|---|---|---|---|---|---|---|---|---|
| Dataset | | Trec | Agnews | Trec | Agnews | IMDB | Chn | Agnews | IMDB | Chn | Trec | | IMDB | | Chn | | Agnews | |
| Noise Ratio | | 40 | | 40 | | | | 40 | | | 20 | 40 | 20 | 40 | 20 | 40 | 20 | 40 |
| BERT-FT | Best | 94.12 | 92.68 | 90.96 | 92.80 | 84.85 | 81.88 | 75.84 | 72.16 | 73.66 | 95.92 | 93.28 | 88.87 | 81.28 | 91.86 | 83.06 | 92.09 | 90.87 |
| | Last | 87.40 | 80.92 | 76.60 | 71.30 | 64.07 | 66.82 | 69.26 | 68.67 | 67.03 | 95.16 | 85.64 | 85.79 | 72.46 | 85.63 | 69.68 | 90.78 | 85.44 |
| CT | Best | 94.16 | 92.98 | 90.20 | 92.61 | 81.35 | 91.21 | 77.34 | 72.72 | 74.06 | 96.16 | 94.20 | 88.30 | 80.87 | 92.91 | 85.01 | 92.37 | 91.23 |
| | Last | 88.28 | 85.75 | 77.04 | 91.17 | 61.85 | 66.58 | 70.01 | 70.09 | 69.06 | 94.88 | 88.68 | 86.22 | 72.68 | 87.53 | 68.95 | 91.21 | 88.21 |
| ELR | Best | 94.68 | 93.06 | 92.64 | 92.88 | 85.70 | 91.83 | 75.65 | 72.08 | 73.57 | 96.24 | **94.64** | 88.92 | 80.58 | 92.65 | 83.63 | 92.30 | 90.94 |
| | Last | 93.48 | 91.68 | 85.24 | 90.73 | 78.28 | 84.91 | 70.18 | 68.77 | 69.01 | 96.16 | 93.52 | 87.15 | 76.10 | 90.28 | 79.58 | 91.74 | 90.11 |
| SelfMix* | Best | 94.04 | 92.91 | 95.32 | 93.22 | 87.41 | 89.02 | 84.09 | **80.74** | 84.02 | 95.64 | 94.16 | **89.88** | **84.34** | 92.91 | 87.70 | 92.11 | 91.15 |
| | Last | 93.56 | 92.71 | 94.96 | 92.95 | 86.19 | 83.45 | 83.15 | 79.99 | 83.52 | 94.28 | 93.60 | 87.88 | 82.67 | 92.03 | 86.66 | 86.55 | 90.24 |
| Ours | Best | **94.72** | **93.27** | **96.32** | **93.56** | **89.13** | **92.65** | **85.40** | 80.07 | **84.25** | **96.48** | 94.36 | 89.05 | 83.47 | **93.23** | **88.47** | **92.65** | **91.33** |
| | Last | 94.20 | 92.33 | 96.00 | 91.06 | 88.21 | 89.99 | 84.76 | 75.42 | 84.03 | 96.16 | 93.68 | 87.87 | 81.58 | 88.36 | 85.42 | 91.74 | 90.25 |

Table 2: Results (%) of five runs on Trec, Agnews, IMDB and Chnsenticorp under different noise settings. The bolded results means the highest among all best scores and the underlined results means the highest among all last scores. Chn is short for Chnsenticorp. * means some hyper-parameters are adjusted according to the type of noise and the composition ratio of mixture noise.

are binary classification datasets, so their symmetric and asymmetric noises are the same.

**Instance-dependent noise:** Follow (Qiao et al., 2022), a LSTM classifier is trained to determine which sample features are likely to be confused. The labels of the samples closest to the decision boundary are flipped to their opposite category based on the classifier prediction probability. All datasets except Trec are used because of its small sample size and uneven categories. Main experiments only show results for a single type of noise with a ratio of 40%, results for other ratios can be found in the Appendix C.

**Mixture of multiple noises:** We mix different types of noise to verify the ability of the algorithm to deal with complex and unknown scenarios. For datasets with only two types of noise, including Trec, IMDB and Chnsenticorp, we mix the two noises evenly. Specifically, Sym and Asym are evenly mixed on Trec, Asym and IDN are evenly mixed on IMDB and Chnsenticorp. For Agnews, we mix three noises unevenly for a larger challenge. The result of evenly mixing of two types of noise on Agnews is in Appendix C.

### 4.2 Baselines

**BERT-FT** (Devlin et al., 2018): the benchmark classification model without special methods, directly trained on noisy data.

**Co-Teaching** (CT for short) (Han et al., 2018): a well-known noise learning method, which maintains two models simultaneously and lets each model select clean samples for the other to train.

**ELR** (Liu et al., 2020): ELR designs a regularization term to prevent the model from memorizing noisy labels by increasing the magnitudes of the coefficients on clean samples to counteract the effect

of wrong samples.

**SelfMix**(Qiao et al., 2022): SelfMix first warms up the model and then uses GMM to separate data to perform semi-supervised self-training, and designs the normalization method according to the characteristics of IDN, resulting in a significant performance improvement compared to other methods lacking specific designs.

For fair comparison, the backbone model of each method is the same, including a pre-trained encoder BERT[1] and a two-layer MLP. The training time of all methods except SelfMix is set to 6 epochs and their hyper-parameters are the same under different noise settings. For SelfMix, we follow the instructions of its paper to set different hyper-parameters for different datasets and noise types. Specially, the warm time for symmetric and asymmetric noise is 2 epochs, the warm time for instance-dependent noise is 1 epoch, and the semi-supervised self-training time is 4 epochs. The average results are run five times on the code base provided by Qiao et al. (2022).

### 4.3 Parameter settings

The same hyper-parameters as baselines remain unchanged, where the maximum sentence length is 256, the learning rate is 1e-5, the drop rate is 0.1, the size of middle layer of MLP is 768 and optimizer is set to Adam. Additionally, the temperature $\tau$ is 0.5, $\alpha$ of Beta distribution is 0.75.

There are some specific hyper-parameters in our method. Specifically, in warm up stage, the interval $s$ of monitoring training set is 1/10 epoch, the converge range $\varepsilon$ is 0.01, the times $\eta$ is 3 and the

---

[1]the weight of BERT for English task is initialized with bert-base-uncased, and the wight for Chinese task is initialized with chinese-bert-wwm-ext of HIT.

fraction $\rho_1$ is 0.1, the batch size is 12. In hybrid training stage, the fraction $\rho_2$ to divide data is 0.2, the threshold $\delta_1$ is 0.9 and $\delta_2$ is 0.4, and the loss weight $\beta$ is 0.3. The batch size of clean set in hybrid training stage is 3 and the batch size of noisy set is 12 to match the ration $\rho_2$ of their numbers.

The duration of the warm up stage varies under different noise settings. The hybrid training stage lasts for 4 epochs to compare with SelfMix. For generality, the parameters used in all cases are same as above.

### 4.4 Main Results

Table 2 demonstrates the main results of our method and baselines.

**Baselines** BERT-FT is highly affected by noise, with its performance and stability decreasing significantly as the complexity of noise increases. CT and ELR demonstrate strengths in handling simple noise scenarios, such as Trec with 40% symmetric noise and a mixture of symmetric and asymmetric noise. However, they perform poorly under more complex noise settings, such as IDN noise. In contrast, SelfMix excels in cases involving IDN due to its design of class-regularization loss and reduced warm-up time. Among all the methods, only SelfMix needs to adjust specific hyper-parameters based on the dataset and noise type.

**Our method** In contrast to the baselines, our method consistently performs well across all noise scenarios. With simple CCN setting, our method exhibits further improvements in performance compared to well-performing baselines like ELR. With IDN setting, our method outperforms general baselines and even surpasses SelfMix in certain cases, such as on Agnews and Chnsenticorp. Moreover, when confronted with mixed noise, our method consistently achieves top or near-top results while maintaining stability. In conclusion, our approach demonstrates superior performance and adaptability in various noise scenarios, making it more practical for real-world scenarios.

### 5 Analysis

In this section, some experiments are designed to make a more comprehensive analysis of our proposed method. The main results are shown in Table 3 and the correspond analyses are as follows. More analyses can be found in Appendix D.

| Dataset | | Trec | | Chnsenticorp | |
| Noise Type | | Asym | | IDN | |
| Method/Ratio | | 20 | 40 | 10 | 40 |
|---|---|---|---|---|---|
| | Clean | 99.20 | 94.89 | 95.71 | 75.57 |
| Ours | Best | 96.68 | 96.32 | 94.75 | 84.25 |
| | Last | 96.36 | 96.00 | 93.24 | 84.03 |
| w/o linear decay fusion | Best | 95.60 | 90.08 | 93.53 | 83.97 |
| | Last | 95.48 | 89.80 | 92.34 | 81.08 |
| w/o correctness statistic | Best | 96.32 | 95.16 | 93.78 | 74.89 |
| | Last | 96.24 | 94.76 | 92.33 | 68.68 |
| w/o mixup | Best | 96.32 | 96.20 | 94.17 | 79.67 |
| | Last | 95.72 | 95.96 | 90.22 | 74.22 |
| w/o r-drop | Best | 96.04 | 95.72 | 93.30 | 76.58 |
| | Last | 95.40 | 95.20 | 75.08 | 64.53 |
| 1 epoch warm-up time | Best | 94.20 | 93.64 | 94.33 | 83.60 |
| | Last | 93.44 | 93.24 | 90.97 | 83.60 |
| 2 epoch warm-up time | Best | 96.36 | 95.36 | 93.80 | 80.65 |
| | Last | 96.16 | 95.12 | 92.55 | 78.20 |
| 3 epoch warm-up time | Best | 96.60 | 96.80 | 93.30 | 77.15 |
| | Last | 96.36 | 96.52 | 90.33 | 75.87 |
| 4 epoch warm-up time | Best | 96.52 | 96.44 | 93.47 | 70.35 |
| | Last | 96.24 | 96.20 | 88.98 | 69.32 |
| | Clean | 98.94 | 92.40 | 94.76 | 73.68 |
| separate data with GMM | Best | 96.16 | 93.28 | 92.82 | 81.05 |
| | Last | 95.68 | 93.16 | 91.58 | 79.50 |

Table 3: Ablation study results in terms of test accuracy (%) on Trec and Chnsenticorp under different noise. The highlighted rows denotes the ratio of correct labels in the separated clean set.

### 5.1 Ablation experiment

To verify that each component contributes to the overall approach, we remove each component separately and check the test accuracy. To remove linear decay fusion, the original labels of clean set are kept and the pseudo-labels of noisy set are generated by the model. For the design of correctness statistic, the relevant parts including early stopping and linear decay fusion are removed and replaced by normal training. The standard cross entropy loss are directly applied on all samples and the corresponding pseudo labels to remove mixup operation. For R-Drop, the KL-divergence term is eliminated. The first part of Table 3 shows each component is helpful to the final performance. Correctness statistic and R-Drop are more critical to complex setting (i.e. 40% IDN), because the model is greatly affected by this noise. And due to the introduction of linear decay fusion, R-Drop becomes more important in the aspect of keeping the model stable.

### 5.2 Analysis of early stopping

Stopping warm up properly is important for adapting to various noise cases and datasets. To verify this view, experiments with different warm-up times are performed. The numbers of war-up epoch are set to {1,2,3, 4} and the results are shown in the

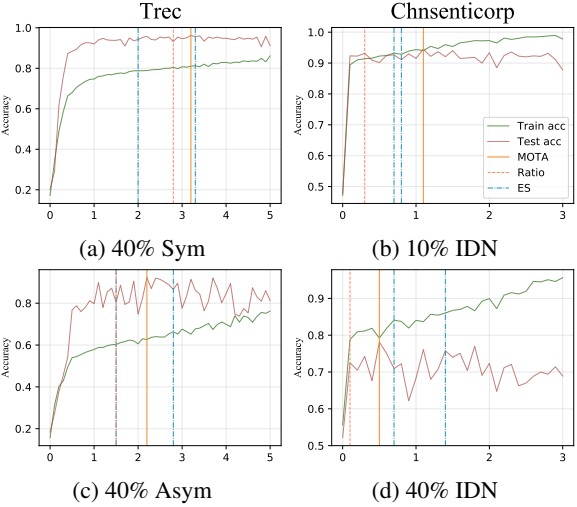

(a) 40% Sym  (b) 10% IDN

(c) 40% Asym  (d) 40% IDN

Figure 4: The instances of early stopping under different noise scenarios.

second part of Table 3. We can observe that no one fixed warm-up time is suitable for all situations. For simple cases such as asymmetric noise, warm up for 3 epochs is beneficial. For complex cases such as instance-dependent noise, warm up for 1 epoch is enough. But adaptive warm-up works better than fixed times on processing all cases. Therefore, finding appropriate stopping point according to different noisy training datasets may be a good strategy of noise learning based on pre-trained models.

To check whether our early stopping method finds the stopping point close to the maximum obtainable test accuracy (MOTA), we draw some finding instances compared with other two heuristic approaches, including stopping through noise ratio (Arpit et al., 2017) and clean validation set (we use test set for validation here). As shown in Figure 4, the estimated start and end points of early stopping (ES) are close to the MOTA under different noise settings. And in some cases ES is closer to the MOTA than stopping through noise ratio (Ratio) especially under IDN. Because IDN is easier fitted by deep models, which resulting in high training accuracy at early stage. Compared with the two methods relying on additional conditions, our method only relies on the original noisy training set, which is more adaptable to the real world.

### 5.3 The design of correctness statistic

During warm up stage, a correctness statistic is maintained to judge the time of early stopping. In addition, it is used to separate the training data into clean set and noisy set for training in hybrid stage. We directly select the top 20% samples of the high-

| Methods\Datasets | Trec | Chnsenticorp | Agnews |
|---|---|---|---|
| BERT-FT | 3min | 22min | 2h 17min |
| SelfMix | 5min | 42min | 4h 40min |
| Ours | 5min | 43min | 4h 48min |

Table 4: The time cost example of different methods under the same noise setting.

est value in the correctness statistic as the clean set. It is different from previous noise-learning works (Li et al., 2020; Qiao et al., 2022; Garg et al., 2021) where the sets are separated by GMM or BMM. To make comparison, GMM is used in hybrid training stage and divide data at each epoch. In addition to the results, the ratios of correctly separation (i.e. how many samples in clean sets are truly clean) are also listed in Table 3, and highlighted with the gray background. As shown, the right ratio of our method is slightly higher than that of GMM, and the performance is better especially under the higher noise ratio settings.

### 5.4 Computational cost

Compared to previous noise learning approaches that include a normal warm-up phase (Han et al., 2018; Qiao et al., 2022), our method involves an additional cost incurred by inferring the entire training set at intervals during the adaptive warm-up phase. This cost is directly proportional to the number of training set samples and the frequency of inference. To strike a balance between monitoring and cost, we set the interval to 1/10 epoch in our experiments. Additionally, during the hybrid training stage, we reduce the cost of inferring the training set and fitting the GMM compared to Self-Mix. Table 4 provides an example of the time costs of BERT-FT, SelfMix and our method under the same settings, with all methods trained on a single GeForce RTX 2080Ti.

### 6 Conclusion

Based on the unknown and complex nature of noise, we propose a noise learning framework based on pre-trained models that can adapt to various textual noise scenarios. This framework automatically stops the warm-up process based on the magnitude and complexity of the noise to prevent the model from overfitting noisy labels. To further leverage mislabeled data, the linear decay fusion strategy is combined with mixup and R-Drop to improve performance while maintaining stability. Experimen-

tal results demonstrate that our method achieves performance comparable to state-of-the-art in all settings within common noise range.

## Limitations

We would like to introduce a versatile noise framework that can adapt to various noise scenarios and have conducted extensive experiments across different simulated scenarios to evaluate the performance. However, it is crucial to acknowledge that we didn't experiment on real textual noise scenarios. If the noise learning method can be verified in real industrial datasets, it will be more convincing. Furthermore, due to the necessity of monitoring the training set during the warm-up stage, the overall training time of our method tends to be longer compared to other approaches, especially when dealing with large datasets like Agnews. Resolving this issue or exploring alternative approaches to reduce training time is a direction that requires further investigation, which we leave to future work.

## Acknowledgements

This work was supported by the Science and technology support program of Sichuan Province under Grant 2022YFG0313.

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

## A Examples of Assumption

The more examples of two assumptions are shown in Figure 5 and 6.

## B Pseudo-code of Adaptive Warm-up

We summarize the pseudo-code of adaptive warm-up in Algorithm 1.

## C Experiments

### C.1 Detailed Results

The detailed results are given in this section. Table 5 shows the results of different ratios of CCN, Table 6 shows the results of different ratios of IDN, and the results of the mixture of two types of noise on Agnews are supplemented in Table 7.

**Class-conditional noise:** With this simple assumption, our proposed method basically achieves the best scores across all noise cases in Table 5. All methods can obtain very good performance under a low noise ratio due to the deep model is robust to simple noise. However, there are still some gains to be made with our approach in some cases such as the results on Trec and Chnsenticorp with 20% asymmetric label noise. Under a high noise ration i.e. 40%, the baselines are all affected to a greater or lesser extent, but our method performs well on different datasets and outperforms all methods. Additionally, we list the results on clean datasets without noise. The performance gap between clean data and noisy data are getting smaller by applying our method.

**Instance-dependent noise:** Table 6 shows that the complexity of IDN results in a substantial decrease in performance for general methods CT and ELR, indicating that they cannot cope well with this noise setting. In these experiments, SelfMix gets more best scores, which is related to its special designs based on noise type such as reducing the warm-up time, designing class-regularization loss and etc. In contrast, our method exhibits strong performance in all cases and achieves scores close to or even better than SelfMix without making assumptions about the type of noise. Particularly noteworthy are the cases where SelfMix underperforms, such as Trec and Chnsenticorp with 10% noise, whereas our method surpasses even the strongest baseline. This further highlights the versatility and effectiveness of our method across different scenarios.

**Mixture of multiple noises:** As shown in Table 7, mixed noise pose challenges, and various baselines exhibit distinct characteristics. BERT-FT is highly impacted by noise, particularly with a significant drop in the Last score. CT and ELR demonstrate their strengths in handling simple noise scenarios, such as a mixture of symmetric and asymmetric noise. On the other hand, SelfMix excels in cases involving IDN. In contrast, our method consistently achieves top or near-top performance while maintaining stability across all scenarios. It offers a more flexible and adaptable solution for unknown noise cases.

In conclusion, our method gets good enough scores under all noise conditions without knowing the noise type, noise ratio or the characteristics of dataset. Besides, the same hyper-parameters are used for all datasets and noise settings. These factors make our approach more practical in real scenarios.

## D More Analysis

### D.1 Analysis of linear decay fusion

In linear decay fusion, we set $\delta_1 = 0.1$ to let the weight of label of cleaner data decay from 1 to 0.1, and set $\delta_2 = 0.6$ to let that of noisier data from 0.6 to 0. To understand the role of two thresholds, Table 8 lists some results of fixing one of them and changing the other. The first three rows shows that $\delta_1$ has little effect on asymmetric noise but limits the performance of IDN. Because IDN is easily fitted by the model, some noisy samples are inevitably assigned to cleaner set. It is more appropriate to reduce the weight to a small value. As can be observed from the last two lines, small value of $\delta_2$ is also detrimental to IDN, because it introduces much noise from noisier set. But large value lets some valuable information to be lost, the trade-off value 0.6 is a good choice. The setting of the two thresholds allows the proposed method to handle complex noise cases and cover possible noise situations, thus making it well applicable to other datasets or the real world.

### D.2 Stability Analysis

In main experiments, the time of hybrid training is set to 4 epochs to fairly compare with SelfMix. The stability of noise learning methods is also an important aspect, and the training time is extended to verify the stability. Since the warm up time is adaptive, we set different hybrid training times.

| Dataset | | Trec | | | | Agnews | | | | IMDB | | Chnsenticorp | |
|---|---|---|---|---|---|---|---|---|---|---|---|---|---|
| Noise Type | | Sym | | Asym | | Sym | | Asym | | Sym/Asym | | Sym/Asym | |
| Method/Ratio | | 20 | 40 | 20 | 40 | 20 | 40 | 20 | 40 | 20 | 40 | 20 | 40 |
| No Noise | | 97.04 | | | | 94.53 | | | | 92.39 | | 96.50 | |
| BERT-FT | Best | 96.36 | 94.12 | 95.88 | 90.96 | 93.85 | 92.68 | 94.06 | 92.80 | 90.46 | 84.85 | 94.08 | 81.88 |
| | Last | 94.92 | 87.40 | 92.36 | 76.60 | 90.01 | 80.92 | 90.78 | 71.30 | 81.57 | 64.07 | 84.58 | 66.82 |
| CT | Best | 96.60 | 94.16 | 96.04 | 90.20 | 94.02 | 92.98 | 94.19 | 92.61 | 90.65 | 81.35 | 94.53 | 91.21 |
| | Last | 95.32 | 88.28 | 94.92 | 77.04 | 91.27 | 85.75 | 93.58 | 91.17 | 84.73 | 61.85 | 87.61 | 66.58 |
| ELR | Best | **96.60** | 94.68 | 96.16 | 92.64 | 93.98 | 93.06 | 94.30 | 92.88 | 90.65 | 85.70 | 94.03 | 91.83 |
| | Last | 96.24 | 93.48 | 95.24 | 85.24 | 93.69 | 91.68 | 93.71 | 90.73 | 89.52 | 78.28 | 92.75 | 84.91 |
| SelfMix | Best | 96.08 | 94.04 | 95.76 | 95.32 | 93.95 | 92.91 | 94.08 | 93.22 | 91.35 | 87.41 | 94.08 | 89.02 |
| | Last | 94.96 | 93.56 | 94.64 | 94.96 | 90.08 | 92.71 | 90.50 | 92.95 | 90.43 | 86.19 | 89.02 | 83.45 |
| Ours | Best | 96.55 | **94.72** | **96.68** | **96.32** | **94.33** | **93.27** | **94.48** | **93.56** | **91.55** | **89.13** | **94.81** | **92.65** |
| | Last | 96.30 | 94.20 | 96.36 | 96.00 | 92.80 | 92.33 | 92.52 | 91.06 | 90.78 | 88.21 | 93.85 | 89.99 |

Table 5: Results (%) of five runs on Trec, Agnews, IMDB and Chnsenticorp under different ratios of class-conditional noise (CCN, inculding symmetric (Sym) and asymmetric (Asym)). The bolded results means the highest among all best scores and the underlined results means the highest among all last scores. The "No Noise" row shows the results of directly fine-tuning in original clean datasets.

| Dataset | | Agnews | | | | IMDB | | | | Chnsenticorp | | | |
|---|---|---|---|---|---|---|---|---|---|---|---|---|---|
| Method/Ratio | | 10 | 20 | 30 | 40 | 10 | 20 | 30 | 40 | 10 | 20 | 30 | 40 |
| BERT-FT | Best | 91.90 | 88.72 | 84.90 | 75.84 | 89.81 | 84.56 | 78.48 | 72.16 | 93.26 | 87.76 | 80.33 | 73.66 |
| | Last | 91.21 | 86.10 | 78.67 | 69.26 | 88.57 | 82.21 | 75.88 | 68.67 | 90.66 | 83.30 | 74.16 | 67.03 |
| CT | Best | 92.02 | 89.08 | 84.52 | 77.34 | 89.82 | 85.86 | 80.38 | 72.72 | 93.90 | 88.63 | 82.03 | 74.06 |
| | Last | 91.33 | 85.88 | 78.26 | 70.01 | 88.84 | 81.85 | 76.13 | 70.09 | 91.56 | 82.63 | 77.15 | 69.06 |
| ELR | Best | 92.05 | 88.92 | 85.04 | 75.65 | 90.24 | 83.70 | 78.14 | 72.08 | 93.75 | 88.55 | 80.71 | 73.57 |
| | Last | 91.21 | 87.10 | 79.85 | 70.18 | 88.32 | 82.08 | 73.51 | 68.77 | 91.15 | 82.76 | 77.90 | 69.01 |
| SelfMix | Best | 91.86 | **90.17** | **88.92** | 84.09 | **90.50** | **87.76** | 83.12 | **80.74** | 93.38 | **91.64** | **88.58** | 84.02 |
| | Last | 89.71 | 88.63 | 87.94 | 83.15 | 88.97 | 85.09 | 80.84 | 79.99 | 91.45 | 88.62 | 85.95 | 83.52 |
| Ours | Best | **92.40** | 89.93 | 87.72 | **85.40** | 90.43 | 86.56 | **83.42** | 80.07 | **94.75** | 90.23 | 87.63 | **84.25** |
| | Last | 90.77 | 88.01 | 83.40 | 84.76 | 90.17 | 84.70 | 78.07 | 75.42 | 93.24 | 88.63 | 87.12 | 84.03 |

Table 6: Results (%) of five runs on Agnews, IMDB and Chnsenticorp under different ratios of instance-denpendent noise (IDN). The bolded results means the highest among all best scores and the underlined results means the highest among all last scores.

| Mixed Type | | 50% Sym+50% Asym | | | | 50% Asym +50% IDN | | | | | | Uneven mixture of three noises | |
|---|---|---|---|---|---|---|---|---|---|---|---|---|---|
| Dataset | | Trec | | Agnews | | IMDB | | Chnsenticorp | | Agnews | | Agnews | |
| Method/Ratio | | 20 | 40 | 20 | 40 | 20 | 40 | 20 | 40 | 20 | 40 | 20 | 40 |
| BERT-FT | Best | 95.92 | 93.28 | 93.96 | 92.83 | 88.87 | 81.28 | 91.86 | 83.06 | 91.72 | 88.12 | 92.09 | 90.87 |
| | Last | 95.16 | 85.64 | 90.20 | 79.81 | 85.79 | 72.46 | 85.63 | 69.68 | 90.26 | 81.88 | 90.78 | 85.44 |
| CT | Best | 96.16 | 94.20 | 94.17 | 92.89 | 88.30 | 80.87 | 92.91 | 85.01 | 91.91 | 88.43 | 92.37 | 91.23 |
| | Last | 94.88 | 88.68 | 90.76 | 82.94 | 86.22 | 72.68 | 87.53 | 68.95 | 90.58 | 82.66 | 91.21 | 88.21 |
| ELR | Best | 96.24 | **94.64** | 94.08 | 93.13 | 88.92 | 80.58 | 92.65 | 83.63 | 91.73 | 88.34 | 92.30 | 90.94 |
| | Last | 96.16 | 93.52 | 93.88 | 91.58 | 87.15 | 76.10 | 90.28 | 79.58 | 90.76 | 86.03 | 91.74 | 90.11 |
| *SelfMix | Best | 95.64 | 94.16 | 93.94 | 93.08 | **89.88** | **84.34** | 92.91 | 87.70 | 91.80 | 88.93 | 92.11 | 91.15 |
| | Last | 94.28 | 93.60 | 90.70 | 92.98 | 87.88 | 82.67 | 92.03 | 86.66 | 85.89 | 88.70 | 86.55 | 90.24 |
| Ours | Best | **96.48** | 94.36 | **94.30** | **93.39** | 89.05 | 83.47 | **93.23** | **88.47** | **92.13** | **89.18** | **92.65** | **91.33** |
| | Last | 96.16 | 93.68 | 93.65 | 91.39 | 87.87 | 81.58 | 88.36 | 85.42 | 90.88 | 85.65 | 91.74 | 90.25 |

Table 7: Results (%) of five runs on Trec, Agnews, IMDB and Chnsenticorp under different ratios of mixed noise. The bolded results means the highest among all best scores and the underlined results means the highest among all last scores. * denotes the best results chosen from different parameter combinations.

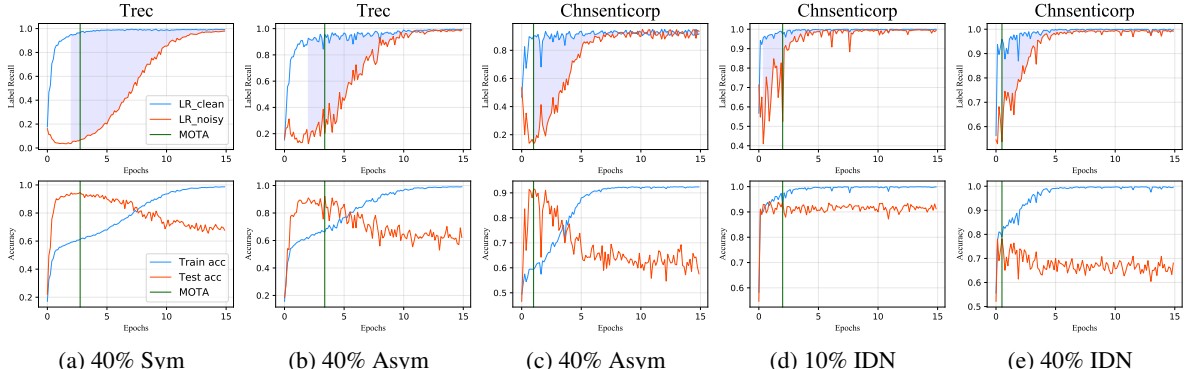

Figure 5: The memorization observation of different noise cases. Label recall on top figures, train and test accuracy on bottom figures.

Figure 6: Examples of model output under different noise settings.

| Dataset | Trec | | Chnsenticorp | |
|---|---|---|---|---|
| Noise Type | Asym | | IDN | |
| Epoch/Ratio | 20 | 40 | 10 | 40 |
| 4   Best | 96.68 | 96.32 | 94.75 | 84.25 |
|     Last | 96.36 | 96.00 | 93.24 | 84.03 |
| 6   Best | 96.80 | 96.64 | 94.40 | 81.98 |
|     Last | 96.36 | 96.04 | 89.72 | 80.68 |
| 8   Best | 97.04 | 96.52 | 94.22 | 83.78 |
|     Last | 96.20 | 91.84 | 89.63 | 81.93 |
| 4+2   Best | 96.84 | 96.44 | 94.75 | 85.37 |
|     Last | 96.48 | 96.12 | 87.43 | 83.15 |
| 4+4   Best | 96.88 | 96.52 | 94.75 | 85.37 |
|     Last | 95.84 | 94.80 | 90.18 | 82.93 |

Table 9: Results in terms of test accuracy (%) with different hybrid training times on Trec and Chnsenticorp under different noise.

There are two modes to extend because of the existence of linear decay fusion. The first is that the decay time is extended as same as the training time, which is expressed as a uniform number in the table. The second is that the decay time is fixed to 4 epochs and the total training time is extended, which is expressed in the form of adding numbers. For example, "4+2" in Table 9 means the weight of original labels decays to the wanted threshold in four epochs and remains unchanged during two more epochs. The results shows that as training time increases, the accuracy of the last epoch decreases but not by much. And interestingly, the Best scores get better in some cases because there are more monitored moments.

### D.3 The trend of correctness statistic

Figure 7 displays the trend of the correctness statistic $ratio^t$ along with the curve representing the

| Dataset | Trec | | Chnsenticorp | |
|---|---|---|---|---|
| Noise Type | Asym | | IDN | |
| Weight Threshold/Ratio | 20 | 40 | 10 | 40 |
| (0.1, 0.6)   Best | 96.68 | 96.32 | 94.75 | 84.25 |
|     Last | 96.36 | 96.00 | 93.24 | 84.03 |
| (0.8, 0.6)   Best | 96.64 | 96.52 | 93.97 | 82.58 |
|     Last | 96.20 | 95.80 | 93.15 | 81.68 |
| (0.5, 0.6)   Best | 96.64 | 96.32 | 94.27 | 81.95 |
|     Last | 96.04 | 95.84 | 93.42 | 79.68 |
| (0.1, 0.2)   Best | 96.64 | 96.24 | 94.10 | 79.67 |
|     Last | 96.12 | 95.76 | 91.52 | 79.37 |
| (0.1, 0.8)   Best | 95.96 | 95.28 | 94.18 | 85.08 |
|     Last | 95.88 | 94.80 | 91.52 | 84.32 |

Table 8: Results in terms of test accuracy (%) with different warm-up times on Trec and Chnsenticorp under different noise.

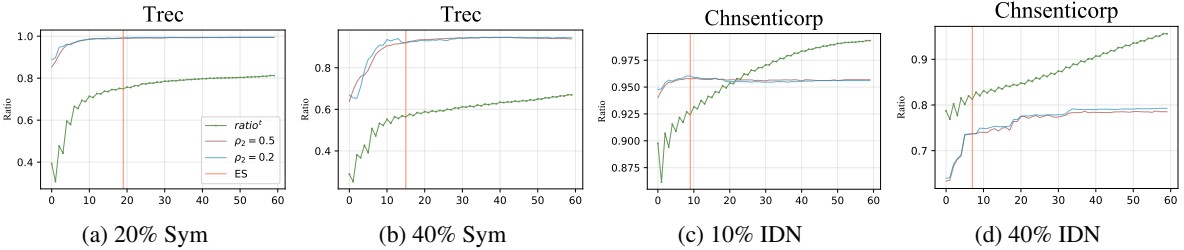

Figure 7: The trend of correctness statistic and the proportion of samples correctly partitioned into the clean set under different $\rho_2$ values.

proportion of samples correctly partitioned into the clean set under different $\rho_2$ values during warm-up. Each point on the correctness statistic curve corresponds to a monitored situation. We observe the appearance of a turning point in various scenarios, and its occurrence time is determined by the speed at which the model fits the training samples. When the turning point appears, the frequency of occurrences within a fixed range quickly rises. Therefore, we set the value of $\eta$ to 3 and the corresponding range $\epsilon$ to 0.01. This is a moderate choice that can accommodate various noise scenarios. A larger $\eta$ is also acceptable but would result in a later early stop time.

For $\rho_2$, we compare two values, 0.5 and 0.2. Under simpler noise settings, such as with 20% symmetric noise, the two $\rho_2$ curves remain relatively flat, indicating that the cleanliness of the top 20% and top 50% sets is similar. However, under more complex settings, such as with 40% idn, $\rho_2 = 0.2$ exhibits some advantages. Additionally, a smaller $\rho_2$ value can accommodate a higher noise ratio.

**Algorithm 1:** Adaptive Warm-up

**Input:** $\theta$ and $\zeta$, training set $(X, \tilde{Y})$, monitoring interval $s$, converge range $\varepsilon$, times $\eta$, accuracy fraction $\rho_1$

Initial: P $= \emptyset$, $enter$ = False ;

**while** $t <$ MaxStep **do**

    Draw a mini-batch $\{(x_b, y_b); b \in (1, ..., B)\}$ from $(X, \tilde{Y})$ ;

    Optimize $\theta$ and $\zeta$ by Equation 1 ;                   `// standard training`

    `/* monitoring the training process at intervals`                    `*/`

    **if** $t \bmod s == 0$ **then**

        Compute $r_i^t$ by Equation 3, $(x_i, \tilde{y}_i) \in (X, \tilde{Y})$ ;

        $m_i^t = \sum_t r_i^t$ ;                  `// update the correctness statistic`

        $ratio^t = \frac{1}{N} \sum_i \mathbb{I}(m_i^t > 0)$ ; `// calculate the proportion of samples that have been fitted`

        P $\leftarrow$ P $\cup \left\{ \left\lceil \frac{ratio^t}{\varepsilon} \right\rceil \right\}$ ;         `// determine the range that` $ratio^t$ `stays inside`

        **if** $not\ enter$ **then**

            **if** P has $\eta$ same items **then**

                $enter$ = True ;                `// enter the NR phase`

                Compute training accuracy $acc$ ;

                $stop\_acc = acc + (1 - acc) \cdot \rho_1$ ;

                                 `// calculate the space available for further warming up`

            **end**

        **else**

            Compute training accuracy $acc$ ;

            **if** $acc \geq stop\_acc$ **then**

                Break ;                   `// stop warming up`

            **end**

        **end**

    **end**

**end**

**Output:** $\theta$ and $\zeta$, $M = \{m_i^t\}_{i=1}^N$ ;       `// output trained model and correctness statistic`