# OpenReview forum: "Adaptive Textual Label Noise Learning based on Pre-trained Models"
_EMNLP/2023/Conference — EMNLP 2023 Findings_

### Official Review · Reviewer_keZc · 2023-07-20

**Paper Topic And Main Contributions:** 1. Proposing an adaptive textual labe…
**Soundness:** 3

**Excitement:**

3: Ambivalent: It has merits (e.g., it reports state-of-the-art results, the idea is nice), but there are key weaknesses (e.g., it describes incremental work), and it can significantly benefit from another round of revision. However, I won't object to accepting it if my co-reviewers champion it.

**Questions For The Authors:**

 I will reconsider my score in the rebuttal

**Reasons To Accept:**

1. The paper addresses a very relevant problem in NLP and Machine Learning, i.e., dealing with label noise. The approach is novel and could be very useful in real-world scenarios where label noise is prevalent.
2. The adaptive warm-up stage with a dynamic early stopping method is a clever idea that prevents overfitting to the noise in the early training process.
3. The hybrid training stage, which incorporates several generalization strategies, is an effective approach to handle the correction of mislabeled instances.
4. The experiments are comprehensive, covering a variety of noise scenarios. The results convincingly demonstrate the robustness and effectiveness of the proposed method.

**Reasons To Reject:**

1. The paper does not discuss the computational cost of their approach in detail. Given that it involves a pre-training phase, the computational requirements could be significant.
2. While the paper claims that the method works well in various noise scenarios, it's not clear if the method is practical for all types of noise. For instance, it might not be effective in scenarios where noise is extreme or highly structured.
3. The paper lacks a deeper analysis of the reasons why their approach performs better than other methods. Some insights into the model's behavior could further enhance the understanding of why their proposed method works well.
4. It's not clear how much the performance of the proposed method depends on the quality of the pre-trained models used. This is an important factor that could influence the applicability of their method in different settings.

**Reproducibility:**

3: Could reproduce the results with some difficulty. The settings of parameters are underspecified or subjectively determined; the training/evaluation data are not widely available.

**Reviewer Confidence:**

4: Quite sure. I tried to check the important points carefully. It's unlikely, though conceivable, that I missed something that should affect my ratings.

---

> ### Author Rebuttal · Authors · 2023-08-29
>
> Thank you for the thorough review and constructive advices. We respond to your comments below:
> **********
> (R1). **The paper does not discuss the computational cost of their approach in detail. Given that it involves a pre-training phase, the computational requirements could be significant.**
> > Our method doesn’t involve a pre-training phase, but rather fine-tuning the pre-trained model which contains the adaptive warm-up stage and hybrid training stage. Indeed, compared to normal warm-up in other approaches like co-teaching (Han et al., 2018) and selfmix (Qiao et al., 2022), adaptive warm-up needs the extra cost of inferring the whole training set at intervals to update the correctness statistic. To save this cost, we set the interval to 1/10 epoch in our experiments. In addition, in hybrid training stage, we save the cost of inferring the training set and fitting the GMM compared to selfmix. We show the time cost of BERT-FT (fine-tuning noisy data directly), selfmix and our method in the same setiting as example:
> Method\Dataset  |    Trec     |  Chnsenticorp  |   Agnews
> ---|:--:|:---:|---:
> BERT-FT             |   3min    |       22min    |        2h17min
> selfmix                  |  5min     |      42min    |        4h40min
> ours                       | 5min      |     43min     |       4h48min
>
> >References:
> Bo Han, Quanming Yao, et al. 2018. Co-teaching: Robust training of deep neural networks with extremely noisy labels.
> Dan Qiao, Chenchen Dai, et al. 2022. SelfMix: Robust learning against textual label noise with self-mixup training.
> **********
> (R2). **While the paper claims that the method works well in various noise scenarios, it's not clear if the method is practical for all types of noise. For instance, it might not be effective in scenarios where noise is extreme or highly structured.**
> >1) We conduct experiments within the noise range of 40% for two reasons: 1) considering that the ratio of corrupted labels in real-world datasets is reported to range from 8.0% to 38.5% (ref1); 2) making consistent comparison with previous work. For extreme scenarios, we conduct experiments with 50%, 60% and 80% label noise on Trec and Chnsenticorp. The performance of baselines and our method drops dramatically as the noise ratio increases. However, our method can still achieve good gains under different types of noises. We show the results on Trec with different level of symmetric noise as example (Best (Last)):
>  Method\Ratio  |    50     |  60  |   80
> ---|:--:|:---:|---:
> BERT-FT             |   61.6 (48.8)    |      42.8 (28.6)   |       31.6 (20.0)
> selfmix                  |  68.8 (54.4)   |       47.8 (34.2)    |      37.0 (30.8)
> ours                       |  69.2 (68.2)    |     48.6 (36.7)    |       37.8 (29.6)
> 2) Structured noise may exist in the generation task whose label is a sentence, but the label of classification task in our focus is a category. In fact, IDN is structured noise to some extent whereby noisy labels are predicted by the same model that has the same bias.
>
> >References:
> Song H, Kim M, et al. 2022. Learning from noisy labels with deep neural networks: A survey.
> *************************
> (R3). **The paper lacks a deeper analysis of the reasons why their approach performs better than other methods. Some insights into the model's behavior could further enhance the understanding of why their proposed method works well.**
> >Thank you for the suggestion. We supplement some analysis details in the paper including: 1) the motivation of introducing early stopping; 2) setting different warm-up time for other approaches that contain the warm-up stage; 3) the learning curves during the whole training phase and the curves of the correctness statistic during warm-up; 4) more analysis about the role of the linear decay fusion strategy under different noise settings.
> *****************************
> (R4). **It's not clear how much the performance of the proposed method depends on the quality of the pre-trained models used. This is an important factor that could influence the applicability of their method in different settings.**
> >In our experiment setting, we use different models for different language task. They both work well and show the same performance trend, which verifies the generalization of our method on pre-trained models. In addition, we compare the performance based on BERT, RoBERTa and ALBERT on Trec to check the dependency of the proposed method. The result show that: 1) The performance and robustness are all improved using different pre-trained models; 2) The performance of the proposed method does depend on the quality of the pre-trained model and shows a positive correlation. For example, the performance of RoBERTa is comparable to that of BERT on BERT-FT and our method, while ALBERT underperforms BERT by 4.84 points on BERT-FT and 4.054 points on our method.

---

### Official Review · Reviewer_fkV9 · 2023-08-05

**Soundness:** 2

**Excitement:**

2: Mediocre: This paper makes marginal contributions (vs non-contemporaneous work), so I would rather not see it in the conference.

**Paper Topic And Main Contributions:**

Adaptive Textual Label Noise Learning based on Pre-trained Models

In this paper the authors describe a pipeline to improve the robustness of learning methods to different types of label noise. They try to identify when a model starts to overfit data due to noise. Based on this they divide the data into "clean" and "noisy" sets. They then apply pseudo labels to both sets with decaying weights, and also generate novel examples with mixup and encourage accuracy with r-drop. They show experimental results on some datasets using synthetically added noise.

**Reasons To Accept:**

+ dealing with noise is an important problem
+ paper is mostly well written

**Reasons To Reject:**

- complex pipeline with numerous ad-hoc parameters and choices
- some aspects such as mixup do not seem well motivated to address the noise case
- experiments are a synthetic setup with added noise of different types and it is not clear how well the approach will work on real data
- many typos and language issues

**Reproducibility:**

3: Could reproduce the results with some difficulty. The settings of parameters are underspecified or subjectively determined; the training/evaluation data are not widely available.

**Reviewer Confidence:**

3: Pretty sure, but there's a chance I missed something. Although I have a good feel for this area in general, I did not carefully check the paper's details, e.g., the math, experimental design, or novelty.

---

> ### Author Rebuttal · Authors · 2023-08-29
>
> Thank you for the review, here are our response to your commonts:
> ******************************
> (R1). **complex pipeline with numerous ad-hoc parameters and choices**
> >1) Our approach first warm up the pre-trained model and then train it by incorporating some generalization strategies. Such training mode is adopted in most noise learning methods such as Han et al.(2018), Jindal et al. (2019), Qiao et al. (2022) .
> 2) Differently, we introduce some hyper-parameters to extend the applicable types and scenarios, but also reduce other parameters for their specific modules like GMM in selfmix (Qiao et al., 2022) and data selection in co-teaching (Han et al., 2018). Specifically, the parameters for determining the early stopping point based on the change trend of the label recall curves, and the parameters for linear decay fusion are set to cover possible noise range. Due to limited space, the details for setting parameters are omitted. We will attach them in the appendix. Lastly, compared to other approaches such as co-teaching, the number of such parameters falls within an acceptable range.
>
> >References:
> Bo Han, Quanming Yao, et al. 2018. Co-teaching: Robust training of deep neural networks with extremely noisy labels.
> Ishan Jindal, Daniel Pressel, et al. 2019. An effective label noise model for DNN text classification.
> Dan Qiao, Chenchen Dai, et al. 2022. SelfMix: Robust learning against textual label noise with self-mixup training.
> *************************
> (R2). **some aspects such as mixup do not seem well motivated to address the noise case**
> >We apologize for not explaining this motivation clearly. In fact, mixup creates more sample points near decision boundaries to make the model capable of dealing with the confused samples in complex noise cases such as IDN. Besides, mixup facilitates mining implicit relations between sentences and learning presentations, which is beneficial to predict accurate pseudo-label as optimize targets. In the ablation study (Sec 5.1), the result of removing the mixup component also shows that it is useful to keep the stability and robustness of the model especially under the 40% IDN.
> **********************
> (R3). **experiments are a synthetic setup with added noise of different types and it is not clear how well the approach will work on real data**
> >We experiment on the synthetic setup for three reasons: 1) it is easier to explore the effects of various noise scenarios on the model to analyze the aspects of the method that need to be improved; 2) for consistent comparison of previous work ; 3) to the best of our knowledge, there is no benchmark for testing label noise in NLP so far. In the future work, we will continue to test our method in real-world classification scenarios.
> *****************************
> (R4). **many typos and language issues**
> >Thank you for giving a thorough review. We have proofread the complete paper carefully and tried to correct all the typos and mistakes. E.g. line 409 “its opposite category” is changed to “their opposite category”, line 462 “remains” is changed to “remain”, line 582 “GMM are used… and to divide …” is changed to “GMM is used … and divide” and etc.

---

### Official Review · Reviewer_SWmB · 2023-08-06

**Soundness:** 3

**Excitement:**

3: Ambivalent: It has merits (e.g., it reports state-of-the-art results, the idea is nice), but there are key weaknesses (e.g., it describes incremental work), and it can significantly benefit from another round of revision. However, I won't object to accepting it if my co-reviewers champion it.

**Paper Topic And Main Contributions:**

The paper presents an adaptive textual label noise learning framework to address the challenge of unpredictable and mixed label noise in real-world scenarios. The framework uses pre-trained models and consists of two stages: an adaptive warm-up stage and a hybrid training stage. The warm-up stage employs an early stopping method based on the training set to dynamically terminate the process according to the model's fit to different noise scenarios. The hybrid training stage incorporates generalization strategies to gradually correct mislabeled instances, effectively utilizing noisy data. Experimental results on multiple datasets demonstrate that the approach outperforms or performs comparably to state-of-the-art methods, even in scenarios with mixed types of noise.

**Reasons To Accept:**

The learning with noise is a widely used setting and very important problem.   The paper introduces an adaptive textual label noise learning framework, which is a novel approach to handling unpredictable and mixed label noise in real-world scenarios.  The adaptive warm-up stage with an early stopping method is a key contribution of the paper. It dynamically terminates the warm-up process based on the model's fit to different noise scenarios, ensuring efficient and effective utilization of resources. The authors propose several assumption and provide a discussion ground and form deep insights of noise learning setting.

**Reasons To Reject:**

1. The experiments and study are limited to classification setting, where most of the datasets are relatively easy. The experimental setting raise some questions (1) If some of assumptions only hold when a learning task is easy? E.g., clean labels are easy to learn compared to incorrect labels. The RTE task and CoLA task are relatively difficult compared to current tasks, does the assumption still hold true?

2. Given the first assumption, there may be different cases: the clean labels may not be fast to learn compared to incorrect labels consider that the data difficulty level is different. Additionally, considering that annotation errors are more likely to occur in challenging tasks, employing early stopping based on this assumption may lead to the learning process not adequately covering these intricate cases. The label noises are synthetically generated, posing a potential risk of introducing bias that deviates from real-world settings.


**Reproducibility:**

3: Could reproduce the results with some difficulty. The settings of parameters are underspecified or subjectively determined; the training/evaluation data are not widely available.

**Reviewer Confidence:**

3: Pretty sure, but there's a chance I missed something. Although I have a good feel for this area in general, I did not carefully check the paper's details, e.g., the math, experimental design, or novelty.

---

> ### Author Rebuttal · Authors · 2023-08-29
>
> Thank you for recognizing the significance of our work. We respond to your comments below:
> ***************************
> (R1). **The experiments and study are limited to classification setting, where most of the datasets are relatively easy. The experimental setting raise some questions (1) If some of assumptions only hold when a learning task is easy? E.g., clean labels are easy to learn compared to incorrect labels. The RTE task and CoLA task are relatively difficult compared to current tasks, does the assumption still hold true?**
> >1) We focus on classification task because there is an obvious distinction between classification and generation tasks in addressing label noise. Although classifying such datasets without label noise is straightforward for current pre-trained models, achieving accurate classification becomes challenging in the presence of label noise. For example, BERT-FT (fine-tuning noisy data directly) obtains 97.04% on Agnews without noise while only 75.84% with 40% IDN.
> 2) Yes, the assumption still hold true in these relatively difficult tasks. We download RTE dataset and corrupt its labels as depicted in our paper to verify it. In the early phase, the label recall rate of clean samples still reaches a higher level than that of erroneous samples, although this trend is somewhat smoother compared to experimental datasets.
> ******************************************************
> (R2). **Given the first assumption, there may be different cases: the clean labels may not be fast to learn compared to incorrect labels consider that the data difficulty level is different. Additionally, considering that annotation errors are more likely to occur in challenging tasks, employing early stopping based on this assumption may lead to the learning process not adequately covering these intricate cases. The label noises are synthetically generated, posing a potential risk of introducing bias that deviates from real-world settings.**
> >1) Indeed, difficult samples do require more time to learn. However, in the presence of label noise, most clean labels are faster to learn than erroneous ones. Moreover, for the more complex samples that were not learned during the warm-up stage, the subsequent hybrid training stage continues to learn them along with the noisy samples.
> 2) We experiment on the synthetic setup for three reasons: 1) it is easier to explore the effects of various noise scenarios on the model to analyze the aspects of the method that need to be improved; 2) for consistent comparison of previous work ; 3) to the best of our knowledge, there is no benchmark for testing label noise in NLP so far. In the future work, we will continue to test our method in real-world classification scenarios.

---

### Meta-Review · Area_Chair_uk4Y · 2023-09-17

**Recommendation:** 3

**Metareview:**

This paper introduces an adaptive textual label noise learning framework, incorporating both an adaptive warm-up stage and a hybrid training stage that utilizes pre-trained models to address label noise. Additionally, it presents a dynamic early stopping method for the warm-up phase, evaluates the model's adaptability to various noise scenarios, and integrates generalization strategies in the hybrid training stage to optimize the utilization of noisy data. Experimental evidence demonstrates the framework's robust performance across different noise scenarios, including those with a noise range of up to 40% and mixed noise types.

There is a consensus among reviewers regarding the significance of the label noise problem in the fields of NLP and ML. Most reviews emphasize that the experiments demonstrate the robustness and effectiveness of the proposed method, particularly in synthetic noise scenarios. However, some concerns center around the simplicity of the tasks and the nonapparent computational complexity analysis of the method. It would be beneficial for the authors to offer further clarification regarding these computational complexities and the potential challenges stemming from the multitude of ad-hoc parameters and choices in future versions of the paper. Addressing these points would enhance the paper's applicability to real-world scenarios where label noise is prevalent and facilitate future iterations of this research.

---

### Decision · Program_Chairs · 2023-10-07

**Decision:**

Accept-Findings

**Comment:**

This paper introduces an adaptive textual label noise learning framework, incorporating both an adaptive warm-up stage and a hybrid training stage that utilizes pre-trained models to address label noise. Additionally, it presents a dynamic early stopping method for the warm-up phase, evaluates the model's adaptability to various noise scenarios, and integrates generalization strategies in the hybrid training stage to optimize the utilization of noisy data. Experimental evidence demonstrates the framework's robust performance across different noise scenarios, including those with a noise range of up to 40% and mixed noise types.

There is a consensus among reviewers regarding the significance of the label noise problem in the fields of NLP and ML. Most reviews emphasize that the experiments demonstrate the robustness and effectiveness of the proposed method, particularly in synthetic noise scenarios. However, some concerns center around the simplicity of the tasks and the nonapparent computational complexity analysis of the method. It would be beneficial for the authors to offer further clarification regarding these computational complexities and the potential challenges stemming from the multitude of ad-hoc parameters and choices in future versions of the paper. Addressing these points would enhance the paper's applicability to real-world scenarios where label noise is prevalent and facilitate future iterations of this research.